# Serum Levels and Removal by Haemodialysis and Haemodiafiltration of Tryptophan-Derived Uremic Toxins in ESKD Patients

**DOI:** 10.3390/ijms21041522

**Published:** 2020-02-23

**Authors:** Joosep Paats, Annika Adoberg, Jürgen Arund, Annemieke Dhondt, Anders Fernström, Ivo Fridolin, Griet Glorieux, Liisi Leis, Merike Luman, Emilio Gonzalez-Parra, Vanessa Maria Perez-Gomez, Kristjan Pilt, Didier Sanchez-Ospina, Mårten Segelmark, Fredrik Uhlin, Alberto Arduan Ortiz

**Affiliations:** 1Department of Health Technologies, Tallinn University of Technology, 19086 Tallinn, Estonia; Jurgen.Arund@taltech.ee (J.A.); Merike.Luman@regionaalhaigla.ee (M.L.); Kristjan.Pilt@taltech.ee (K.P.); Fredrik.Uhlin@regionostergotland.se (F.U.); 2Centre of Nephrology, North Estonia Medical Centre, 13419 Tallinn, Estonia, Liisi.Leis@regionaalhaigla.ee (L.L.); 3Nephrology Division, Ghent University Hospital, 9000 Ghent, Belgium; Annemie.Dhondt@UGent.be (A.D.); Griet.Glorieux@UGent.be (G.G.); 4Department of Nephrology and Department of Medicine and Health Science, Linköping University, 58185 Linköping, Sweden; Anders.Fernstrom@regionostergotland.se (A.F.); Marten.Segelmark@liu.se (M.S.); 5Fundación Jiménez Díaz University Hospital Health Research Institute, 28040 Madrid, Spain; egparra@fjd.es (E.G.-P.); MVanessa@fjd.es (V.M.P.-G.); sliverco41@gmail.com (D.S.-O.); AOrtiz@fjd.es (A.A.O.)

**Keywords:** uremic toxins, tryptophan, tryptophan-derived uremic toxins, indoxyl sulfate, indole-3 acetic acid, end-stage kidney disease, chronic kidney disease, haemodialysis, haemodiafiltration

## Abstract

Tryptophan is an essential dietary amino acid that originates uremic toxins that contribute to end-stage kidney disease (ESKD) patient outcomes. We evaluated serum levels and removal during haemodialysis and haemodiafiltration of tryptophan and tryptophan-derived uremic toxins, indoxyl sulfate (IS) and indole acetic acid (IAA), in ESKD patients in different dialysis treatment settings. This prospective multicentre study in four European dialysis centres enrolled 78 patients with ESKD. Blood and spent dialysate samples obtained during dialysis were analysed with high-performance liquid chromatography to assess uremic solutes, their reduction ratio (RR) and total removed solute (TRS). Mean free serum tryptophan and IS concentrations increased, and concentration of IAA decreased over pre-dialysis levels (67%, 49%, −0.8%, respectively) during the first hour of dialysis. While mean serum total urea, IS and IAA concentrations decreased during dialysis (−72%, −39%, −43%, respectively), serum tryptophan levels increased, resulting in negative RR (−8%) towards the end of the dialysis session (*p* < 0.001), despite remarkable Trp losses in dialysate. RR and TRS values based on serum (total, free) and dialysate solute concentrations were lower for conventional low-flux dialysis (*p* < 0.001). High-efficiency haemodiafiltration resulted in 80% higher Trp losses than conventional low-flux dialysis, despite similar neutral Trp RR values. In conclusion, serum Trp concentrations and RR behave differently from uremic solutes IS, IAA and urea and Trp RR did not reflect dialysis Trp losses. Conventional low-flux dialysis may not adequately clear Trp-related uremic toxins while high efficiency haemodiafiltration increased Trp losses.

## 1. Introduction

Tryptophan (Trp) is an essential amino acid that is obtained through diet. However dietary Trp uptake is among the lowest for amino acids [1]. Besides contributing to proteins structure, Trp is the only amino acid that binds to a protein—albumin [2,3,4]. Thus, serum includes both albumin-bound Trp and free Trp, with the latter pool being available for tissue uptake [2,4]. Trp is mainly used for protein synthesis, but also for synthesis of important bioactive molecules such as kynurenine, serotonin, tryptamine, and melatonin [5]. In contrast to other amino acids, healthy kidney excretion of Trp is negligible (around 0.5% of creatinine clearance), leading to quite stable serum Trp concentrations [1,6,7]. While an oral Trp overload may transiently increase serum Trp, this does not result in increased kidney excretion of Trp [6,8]. Rather, Trp metabolites are increased in urine.

Ingested Trp is metabolized by gut microbiota to kynurenine, serotonin, and indole with its derivates [9]. Indole derivates from the gut microbiota include several solutes classified as uremic toxins, such as indoxyl sulfate (IS) and indole acetic acid (IAA), based on their biological effects and associations with outcome in ESKD patients [10]. ESKD is usually treated with haemodialysis. However, haemodialysis does not selectively remove toxic uremic solutes, it removes also neutral and necessary solutes, including amino acids [11] such as Trp [12,13,14,15,16,17], potentially contributing to the negative impact of haemodialysis. In addition, uraemia-associated disruption of biochemical homeostasis (e.g., low concentrations of Trp due to malnutrition and removal by haemodialysis), are also suspected to negatively contribute to patient well-being [18,19,20]. 

Haemodialysis has been reported to be a low-efficient method to remove protein-bound uremic toxins, as only free solute is available for removal [21]. Protein-bound uremic toxins include Trp metabolites and the albumin-bound indole derivates IS and IAA [22,23,24]. Both solutes are widely acknowledged as uremic toxins that impair biological pathways and patient outcome [10,25,26]. To our knowledge, there are no published studies reporting both total and free uremic serum levels, removal and rebound-effect of Trp combined with indolic protein-bound uremic toxins, and the multicentre haemodialysis studies are scarce [26,27,28,29,30,31]. 

The aim of this multicentre clinical study was to evaluate serum levels and removal during haemodialysis and haemodiafiltration of Trp and Trp-derived uremic toxins, IS and IAA, in ESKD patients.

## 2. Results

A total of 78 haemodialysis patients were studied. As detailed in “Material and Methods”, 78% were male and mean age was 63 ± 16 years. Table 1 presents the total and free pre-dialysis solute concentrations by centre for all dialysis sessions.

The pre-dialysis levels of total and free IAA are comparable among the four different dialysis centres, being significantly lower than levels of IS. Pre-dialysis Trp, IS and urea levels seem to have rather nonsystematic differences between the centres. Centre 2 had the highest levels of Trp and IS. The largest differences correspond to free Trp levels. However, free Trp levels were similar in patients dialysed in centre 1 and in centre 2. In general, the free:total Trp ratio (mean 0.23 ± 0.08) was higher than the 0.15 previously reported for healthy populations [8].

Mean total and free serum concentrations and total dialysate concentrations of Trp, IS, IAA and urea at different time points during the dialysis are shown in Figure 1. During dialysis, total and dialysate IS, IAA and urea levels decrease. Contrary to other uremic solutes, mean total Trp concentration increased towards the end of the session (*p* < 0.001). Interestingly, the mean free serum Trp and IS concentrations increased during the first hour of the dialysis sessions compared to the initial pre-dialysis levels (Figure 1b) (*p* < 0.001). Generally, dialysate values (Figure 1c) followed the total serum concentration pattern.

The mean total and free serum and dialysate Trp, IS, IAA and urea concentrations normalized for the baseline pre-dialysis value at different time points during the dialysis sessions are shown in Appendix A. The normalized concentrations show similar trends to absolute values shown in Figure 1, however the increase of free serum Trp and IS during the first hour of the dialysis sessions compared to the initial pre-dialysis levels is accentuated.

Reduction ratios (RR) at different time points during the dialysis sessions for Trp, IS, IAA and urea were calculated for total and free serum concentration (Figure 2). The mean RR for Trp behaved significantly different from other uremic solutes. The variability in RR was largest for total and free Trp. As expected, urea RR were higher than RR for Trp and its derivatives. The rebound effect based on the 30 min post-dialysis serum value for the different solutes significantly affected the RR value (*p* < 0.001 and *p* < 0.05 for Trp).

RR at different time points during the dialysis sessions for Trp, IS, IAA and urea calculated based on the dialysate concentrations are shown on Figure 3.

The mean RRs for Trp are significantly different from other uremic solutes. The trends in RR based on dialysate values are similar to RRs based on total and free serum concentrations, except for negative displacer effect at 60 min for free Trp and free IS. For urea, which is not bound to protein, RRs at 240 min, calculated based on the dialysate values, are similar to RR based on serum concentration.

Figure 4 illustrates RRs at 240 min (Figure 4a–c) and total solute removal of urea, Trp, IS and IAA calculated as average over all dialysis sessions (Figure 4d) for different dialysis modalities and settings.

RR and TRS values for urea, IS and IAA for different dialysis modalities and settings behave similarly (Figure 4): mean RR and TRS values are significantly lower for the conventional low-flux dialysis than for other dialysis settings (*p* < 0.001). However, Trp RR values were close to, equal or even below zero and especially the latter two are not representative of real removal of Trp measured by TRS. TRS of Trp was remarkable, given that urine excretion in healthy subjects is negligible, and was very responsive to dialysis modality, with losses being 80% higher on highHDF than on lowHD (see Table 3 for dialysis settings).

## 3. Discussion

This multicentre clinical study reports the total and free serum levels, removal by haemodialysis and haemodiafiltration and rebound-effect of Trp and indolic protein-bound uremic toxins, IS and IAA, in ESKD patients. The main findings of the study were: (1) Pre-dialysis serum Trp, IS and urea levels had nonsystematic variability but there was no statistically significant difference in pre-dialytic total and free serum IAA levels of ESKD patients from participating clinical centres. (2) Total serum and dialysate Trp do not decrease at the end of haemodialysis, and may even increase, contrary to uremic solutes. This results in negative RR values for the total serum Trp or almost constant RR values for dialysate Trp at the end of treatment, respectively, which are not representative of real removal of Trp as assessed by TRS. (3) In this regard, Trp removal was remarkable and very responsive to dialysis modality. (4) Free serum Trp and IS concentrations increase over pre-dialysis levels during the first hour of dialysis sessions. (5) The RR values calculated from dialysate values reflect RR trends for total and free urea, IS and IAA calculated from serum samples.

Standard dialysis regimes applied to different populations with potentially different dietary habits, genetic background, microbiota or even standards of care in different centres may explain the differences in baseline pre-dialysis Trp, IS and urea levels between centres. These differences emphasize the need to perform multicentre studies including geographically dispersed and culturally and genetically different centres. Subjects with residual renal function excrete Trp metabolites although in healthy subjects urinary Trp excretion is negligible and in dialysis patients only 0.3% of dietary Trp intake was excreted in urine [6,12]. In healthy subjects, an oral Trp overload leads to a transient increase in serum Trp [6]. However, earlier studies did not find any associations between plasma Trp concentration and Trp intake in dialysis patients, suggesting a key role for gut microbiota and liver in metabolizing dietary Trp [12,32]. Since our study enrolled participants from different centres across Europe, different dietary habits and differences in gut microbiota can be considered. It is noteworthy that the highest levels of total Trp and IS were observed in the centre with highest total serum protein levels, although free Trp was not higher.

Dietary Trp is metabolized by gut bacteria and the liver and incorporated into proteins and is excreted in urine as Trp metabolites. Gut bacteria Trp metabolism leads to the production of indole and IAA, which behave as or are precursors of uremic toxins [9,33]. Indole biosynthesis from dietary Trp is catalysed by a single-step reaction involving a specific Trpase (EC 4.1.99.1) [34]. The key enzyme for the production of IAA is indolepyruvate decarboxylase (EC 4.1.1.74), which catalyses the conversion of indolepyruvate to indole-3-acetaldehyde [35]. Trp, indole and IAA are absorbed into the circulation where indole is metabolised at the liver to IS and indoxyl glucuronide, while Trp is further metabolised mostly to kynurenine, but in small fraction also to the important neurotransmitter serotonin in the brain [7]. Trp metabolism is regulated by plasma free Trp levels [36]. In the present study, total serum Trp levels were comparable to recent reports in dialysis patients and lower than in healthy populations [12]. In this regard, plasma Trp levels are nearly 60% lower at stage 5 CKD than at stage 1 [37]. Interestingly, low serum Trp has been associated with adverse cardiovascular outcomes and mortality in the general population [38,39] and dialysis patients are at an increased risk of cardiovascular disease and death.

Trp depletion is common in haemodialysis patients. Factors potentially contributing to Trp depletion are reduced dietary Trp intake; increased catabolism due to an inflammatory status and losses during dialysis. Currently, little is known about Trp homeostasis in haemodialysis patients [12]. Although we do not have data on Trp intake, Trp intake was reported to be well above the dietary recommendations in Dutch haemodialysis patients and only 5% of dietary Trp intake was removed by haemodialysis when normalized for 24 h. While most Trp is metabolized and excreted as metabolites, haemodialysis Trp removal is unregulated and well above kidney excretion and takes place within a brief time frame. Thus, it was recently reported that 10% of dietary Trp was removed in a single haemodialysis session [12]. We now report that this may be influenced by dialysis modality, and, assuming similar dietary Trp as previously reported for haemodialysis patients, Trp removal during a single high efficiency HDF session may reach 16% of 24 h dietary intake.

We now confirm in a larger study of more heterogeneous patients that the high Trp removal rate during haemodialysis is not accompanied by decreased serum Trp levels. Indeed, mean total serum Trp increased during dialysis. The cause of this phenomenon is currently unknown [12]. An association between increased plasma Trp during the haemodialysis session and a lower risk of mortality was found in a small study of 40 patients [12]. While the authors speculated that increased serum Trp during dialysis appears to be protective, causality is difficult to assign in observational studies. Thus, increase in serum Trp during dialysis may represent a strong impact of the dialysis procedure and/or Trp removal during dialysis that drive profound changes in Trp generation, metabolism and/or transport across membranes. Thus, in contrast to indole and IAA, which only originate from the intestine, Trp may be ingested or generated endogenously from muscle proteolysis and released into the circulation [40,41]. In this regard, an alternative hypothesis is that the phenomenon is not necessarily protective, but may identify individuals capable of mounting such a response and these may be the healthier individuals.

It can be speculated whether dialysis itself may modify Trp binding to albumin, perhaps causing a shift or binding competition between Trp and its indolic metabolites, similarly to that described for uremic serum 40 years ago [42]. In this regard, an early (60 min) increase in free Trp is observed but it is not maintained during the full dialysis session. The early increase in free solute was also observed for IS. The reason could be a modified albumin binding capacity during dialysis due to administered heparin [43]. Thus, heparin provokes the release of lipolytic enzymes [44], leading to hydrolysis of triglycerides and release of free fatty acid and increasing their plasma concentrations. It has been shown that free fatty acids displace drugs and ligands (also for IS, IAA and Trp) from their protein binding sites [45,46] and that Trp increases displacement of IS and IAA [47]. IS, IAA and Trp primarily bind to Sudlow site II on albumin, with IS having the highest affinity [47,48,49,50]. The effect may differ between patients due to different plasma levels of ligands and uremic toxins that compete for the same binding site on albumin.

The RR and TRS values behaved similarly for different dialysis modalities and settings for urea, IS and IAA as mean RR and TRS values were significantly lower for conventional low-flux dialysis than for other dialysis settings for all solutes (*p* < 0.001), which is in good relation with results of the previous studies [51,52]. However, Trp RR values were not influenced by dialysis modality. Thus, despite Trp being removed by dialysis, as TRS was well over zero, Trp RR values remained close to zero or even negative with all dialysis modalities. In different studies, the same has been noted [12,42,53,54]. This means that to assess Trp removal, only TRS provides an adequate estimation, while serum Trp levels and Trp RR do not.

The major strength of this study is the prospective, multicentre and interventional design testing a larger and more diverse sample size than prior studies. Nevertheless, some limitations should be acknowledged. Thus, dietary Trp data were not available and enrolled patients and dialysis centres may not be fully representative of the countries represented. Additionally, no attempts were made at displacing uremic solutes from proteins by administering displacers to patients, since this does not represent routine clinical care.

The Trp findings should be put in context with the behaviour of other amino acids during haemodialysis and haemodiafiltration. Haemodialysis patients have lower plasma total and nonessential amino acid concentrations than controls and similar essential and branched amino acid concentrations [55]. Loss of amino acids during haemodialysis is thought to potentially contribute to malnutrition. Thus, in a recent study, a single haemodialysis session resulted in the loss of 11.95 ± 0.69 g amino acids via the dialysate, of which 8.26 ± 0.46 g were nonessential amino acids, 3.69 ± 0.31 g were essential amino acids, and 1.64 ± 0.17 g were branched-chain amino acids. As a consequence, plasma total and essential amino acids concentrations declined significantly from 2.88 ± 0.15 and 0.80 ± 0.05 mmol/L to 2.27 ± 0.11 and 0.66 ± 0.05 mmol/L, respectively [56]. Haemodiafiltration increased total amino acid losses by 2.5% versus high efficiency haemodialysis and this was statistically significant [55]. In acute kidney injury, additional modalities of renal replacement therapy are used that increase amino acid loses. Thus, continuous veno-venous hemofiltration resulted in threefold higher amino acid losses that haemodialysis and sustained low-efficiency diafiltration in 50% higher losses than haemodialysis [13]. 

In conclusion, in a multicentre study encompassing patients from diverse European countries, serum Trp concentrations and RR during haemodialysis behaved differently from uremic solutes IS, IAA and urea, and Trp RR did not reflect dialysis Trp losses. Trp losses during dialysis may represent up to 16% of daily intake and occur in the course of a few hours. This appears to trigger adaptive mechanisms that maintain serum Trp levels and, thus, aggravate dialysis Trp losses. Since the response of serum Trp during dialysis has been associated to outcomes in a prior study [12], unravelling the molecular mechanisms involved in maintaining serum Trp levels during dialysis may provide further insights into ESKD pathophysiology. In this regard, we will follow the present prospective cohort long-term to assess the impact of the findings on outcomes. As additional findings, conventional low-flux dialysis may not adequately clear Trp-related uremic toxins while high efficiency haemodiafiltration increased Trp losses. The potential consequences of this differential impact of dialysis modalities on Trp removal merits further studies.

## 4. Materials and Methods 

Clinical study data was acquired from four separate dialysis centres from countries with diverse life expectancy, renal replacement therapy incidence and kidney transplant rates: (1) centre 1: North Estonia Medical Centre, in Tallinn, Estonia (22 patients), (2) centre 2: Linköping University Hospital, in Linköping, Sweden, (21 patients) (3) centre 3: Ghent University Hospital, in Ghent, Belgium (15 patients). (4) centre 4: Fundación Jiménez Díaz University Hospital Health Research Institute, in Madrid, Spain (20 patients). All studies were performed after approval of the protocol by local ethics committees (Tallinn Medical Research Ethics Committee at the National Institute for Health Development, Estonia, decision no. 2205 (issued 27 Dec. 2017); Linköping Regional Medical Research Ethics Committee, Linköping, Sweden, decision no. 2017/593-31 (issued 17 Jan. 2018); Ghent University Hospital, Commissie voor Medische Ethiek, Ghent, Belgium, decision no. B670201938627 (issued 15 Feb. 2019); Fundación Jiménez Díaz Clinical Research Ethics Committee, Madrid, Spain, decisions (no. 9/18, issued 8 May 2018) and (no. 13/18, issued 10 Jul 2018) (Figure 5). Informed consent was obtained from all participating patients. The informed consent templates in local languages were developed for all dialysis centres. To assure the confidentiality, accuracy and security of data and data management, all clinical information and biological samples and tissues obtained from the clinical networks were transmitted only after anonymization/blinding.

Inclusion criteria were following: over 18 y.o. patients on chronic hemodialysis, HD procedures for 3.5–4.5 h thrice weekly preferably via AV fistula or graft, blood access capable to manage blood flow of at least 300 mL/min; absence of clinical signs of infection or other active acute clinical complications and an estimated life expectancy over 6 months. Clinical data of the 78 participants monitored for a total of during 320 dialysis procedures is presented in Table 2.

Each patient was observed during four midweek dialysis sessions. In order to vary dialytic removal of urea and Trp derivatives the following dialysis modalities and haemodialysis (HD)-machine settings were applied (Table 3, which also summarizes the dialyzers used): (1) “Standard” dialysis removed uremic toxins on standard settings [HD or haemodiafiltration (HDF)] as previously prescribed for the patient in routine clinical care. This provided a baseline, validated more extensively the current dialysis prescription, and introduced patients more smoothly into the study. (2) “LowHD” dialysis had minimal dialysis settings to provide conditions for minimal uremic toxin removal; (3) “MediumHDF” dialysis had maximum dialyzer surface area and highest dialysate (d) blood (b) flow ratio (Qd/Qb) expected to increase removal of Trp derivatives; (4) “HighHDF” dialysis had maximum dialysis settings in terms of dialyzer surface area and substitution volume and second highest Qd/Qb.

Dialysis machines were Fresenius 5008 for 304 dialyses, Fresenius 4008 for 4 dialyses and Fresenius 6008 for 4 dialyses (Fresenius Medical Care, Germany). 

To compare the efficiency of the various dialysis treatments settings, blood and spent dialysate samples were taken either from blood lines or the dialysate outlet of the dialysis machine, respectively (Figure 6). Blood samples were taken before the start of the dialysis, 60 min after the start, at the end of the dialysis (slow pump method) and 30 min after the end of the dialysis. All blood samples were taken from the arterial line, i.e., before the dialyzer during dialysis session. The dialysis samples from the drain tube of the dialysis machine were taken at 7, 60, 120 and 180 min after the start of the session and, at the end of the session (240 min). In addition, the waste dialysate was collected into a large tank during the whole procedure for calculating the total amount of uremic toxins removed. After the end of the procedure, the dialysate collection tank was weighed, and one sample was taken from it after careful stirring.

All samples collected during the haemodialysis session were divided into two sets (Figure 6): the first set of samples was directly sent to clinical laboratory and the second set of samples was separated for the analytical laboratory analyses (Lab Tallinn University of Technology (TUT)). Prior to transportation or storage in an ultrafreezer, samples were processed as follows: (A) spent dialysate samples for the clinical laboratory were drawn into the Becton Dickinson Vacutainer SST II Advanced 5 mL (Franklin Lakes, NJ, USA) vacuum tubes. For analytical chemistry analysis, spent dialysate samples were transported in 120 mL Becton Dickinson Vacutainer urine collection cups (Franklin Lakes, NJ, USA) in centre 1. In centre 2, centre 3 and centre 4 the spent dialysate samples were placed into 5 mL Brand cryogenic tubes (Brand GMBH + CO KG, Wertheim, Germany) for freezing; (B) serum samples were kept still for 30 min to allow clotting and were centrifuged for 20 min with swing-out buckets at 3000 g, followed by transportation to the clinical lab for the first tube of serum while the second tube of serum was placed into 2 mL cryotubes for freezing; (C) plasma samples were gently shaken and left still until transportation to the clinical laboratory analysis. 

The first sample set was directly taken to the clinical chemistry laboratories for standard analysis in all centres (Synlab Eesti OÜ in Estonia, Clinical Chemistry Laboratory at Linköping University Hospital in Sweden, Clinical Biochemistry Laboratory at Fundación Jiménez Díaz University Hospital Health Research Institute, in Madrid, Spain, and Laboratory of Clinical Chemistry and Clinical Analysis at Ghent University Hospital, in Ghent, Belgium) in order to measure serum and dialysate urea concentration using standardized methods. 

The second sample set was directly sent to the analytical laboratory without freezing in centre 1 and stored in ultrafreezers (−75 °C) until sent to centre 1 in centres 2, 3 and 4. After collection of samples from 30 dialysis sessions, samples were sent to Tallinn in thermally insulated containers that were filled with dry ice to secure that they remain frozen. In Tallinn, they were unfrozen, processed according to the sample preparation methods and analysed in the analytical chemistry laboratory Tallinn University of Technology using high-performance liquid chromatography (HPLC) to measure Trp and its derivatives (IS, IAA). In order to assess serum total and free concentrations of Trp and its derivates, serum samples were prepared in two separate groups as follows: (1) To measure total concentrations of protein-bound solutes, samples in group one were diluted 1:3 with normal saline in Eppendorf Protein LoBind 1.5 mL tubes (Hamburg, Germany), followed by vortexing at 1500 rpm for 1 min, heating for 30 min at 95 °C and rapid cooling in ice-cooled water bath, followed by centrifugation at 30.000 g for 10 min at 37 °C, the resulting supernatant was filtered by centrifugation. (2) To measure unbound (free) concentrations of protein-bound solutes, samples in group two were only filtered by ultracentrifugation. For filtering by centrifugation Sartorius Vivacon 30 kDa cut-off filters (Göttingen, Germany) were used at 14.000 g for 3 h at 37 °C. Prior to filtering by ultracentrifugation cut-off filters were washed through with 400 µL type I ultrapure water (Millipore Synergy UV, Burlington, MA, USA) by centrifugation at 14.000 g for 15 min at 37 °C. One µL of formic acid (Sigma-Adrich, St. Louis, MO, USA) was added to the filtrate in order to stabilize uric acid to undissociated form in filtrate. The filtrate was placed into 200 µL vial inserts which were kept at 1°C until HPLC analysis for a maximum of 72 h. Before HPLC analysis of dialysis samples, 10 µL of formic acid was added to the dialysate in order to solubilize uric acid.

The HPLC system consisted of a quaternary gradient pump unit, a thermostated autosampler, a column oven, a diode array spectrophotometric detector (DAD), and a fluorescence detector (FLD), all Ultimate 3000 Series instruments from Dionex a division of Thermo Scientific company (Sunnyvale, CA, USA). Two continuous columns of Poroshell 120 C18 4.6 x 150 mm with a security guard Poroshell 120 C18 4.6 x 3 mm were from Agilent Instruments (Santa Clara, CA, United States). The eluent was mixed with 0.05 M formic acid adjusted to pH 4.25 with ammonium hydroxide (A), and organic solvent mixture of HPLC grade methanol and HPLC-S grade acetonitrile, both from Honeywell (Charlotte, NC, USA) in ratio of 9:1 with 0.05 M ammonium formiate salt (B). The three-step linear gradient elution program was used with the total flow rate of 0.6 mL/min at the column temperature of 40 °C. The chromatographic data was processed by Chromeleon 7.1 software by Thermo Scientific.

Aqueous calibration standards were prepared from reagents purchased from Sigma-Aldrich (St. Louis, MO, USA) with purity of assay >98% (TLC) diluted in type I ultrapure water (Millipore Synergy UV, Burlington, MA, USA), followed by ultrasonic bath for 30 min to ensure complete dissolution of the reagents. The standard solutions were analysed with the same HPLC method as the serum and the spent dialysate samples. Peak areas were plotted in function of concentrations and all concentrations were expressed as micromoles per litre.

All the results were assessed for possible errors and data conformity. The errors and problems were related with sample drawing, correct sample labelling and clinical laboratory analysis. The 240-min spent dialysate samples were drawn after the end of dialysis at 8 dialysis sessions. As these samples were comparable with clean dialysate, these datapoints were not included to the data analysis. As spent dialysate samples are highly diluted biologic fluids, some analyte concentrations were below quantification limit with clinical laboratory methods, especially in the case of urea. These datapoints were not included to the data analysis. In case the concentration of the analyte exceeded the upper limit of quantification, the analysis was repeated with diluted sample of lower volume of that sample.

Statistical evaluation was performed with MATLAB R2019b (Mathworks, Natick, MA, USA). Parametric statistical tests were used for differences between subjects via an unpaired *t*-test, and for differences between related samples of the same patients via a paired *t*-test. A *p*-value of <0.05 was considered significant.

To assess uremic retention solute removal during a dialysis session the reduction ratio (RR) in percentage was calculated from the solute concentrations as:(1)$$RR=\frac{C_{0}-C_{t}}{C_{0}}100\%$$
where *C*_0_ is a serum sample before and *C*_t_ after dialysis procedure, or at the predetermined time points after start of the dialysis. For spent dialysate, a sample taken after 7 min from the start and at the end of dialysis procedure was utilized as *C*_0_ and *C*_t_, respectively.

The total removed solute (TRS) of a substance was calculated based on total dialysate collection (TDC) as follows:TRS = *C*_T_ × *W*_T_(2)
where *C*_T_ is the substance concentration in total dialysate collection tank (µmol/L) and *W*_T_ is the weight of the dialysate collection tank (kg). It was assumed that average density of spent dialysate is 1008 ± 0.001 kg/L [59].

## Figures and Tables

**Figure 1 ijms-21-01522-f001:**
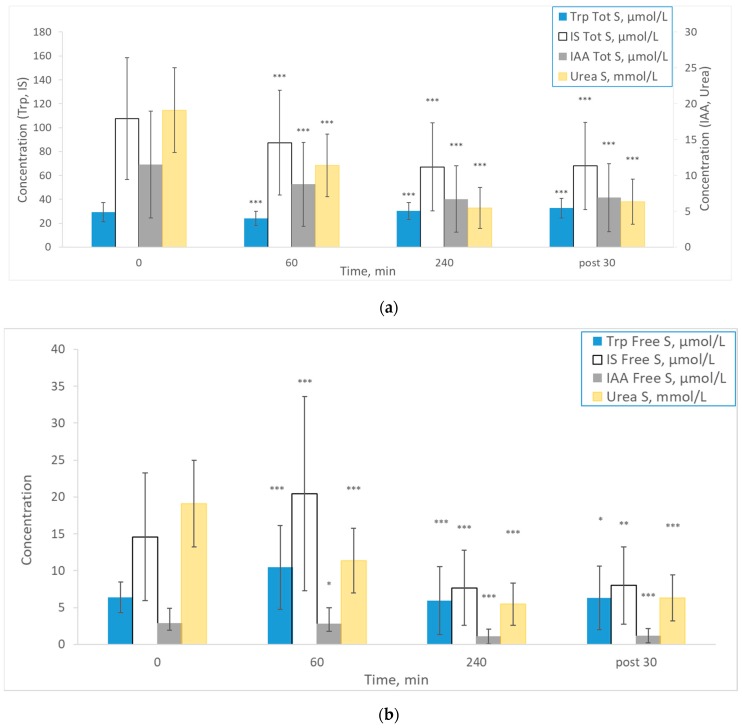
Mean ± SD concentrations at different time points during the dialysis sessions for Trp, IS, IAA (µmol/L) and urea (mmol/L) for (**a**) total serum values (*n* = 310); (**b**) free serum values (*n* = 310); and (**c**) dialysate values (*n* = 257). *** *p* < 0.001; ** *p* < 0.01; * *p* < 0.05 vs. previous timepoint value.

**Figure 2 ijms-21-01522-f002:**
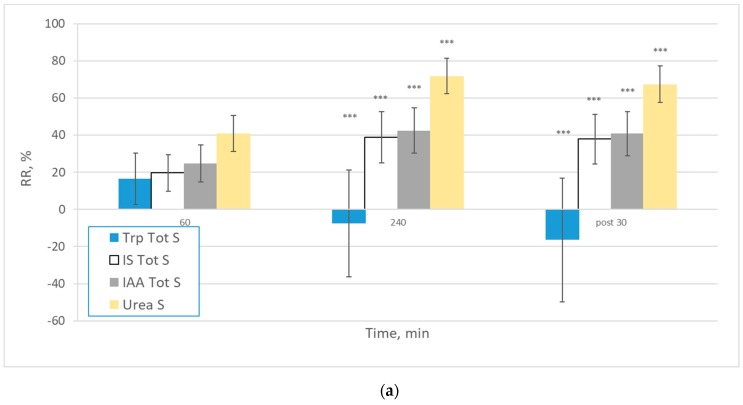
Reduction ratios (RR) at different time points during the dialysis sessions for Trp, IS, IAA and urea calculated from (**a**) total serum values (*n* = 309) and (**b**) free serum values (*n* = 309). *** *p* < 0.001; ** *p* < 0.01; * *p* < 0.05 vs. previous timepoint value.

**Figure 3 ijms-21-01522-f003:**
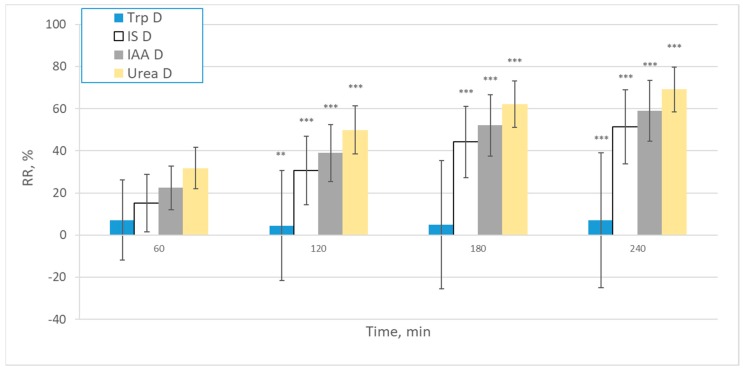
Reduction ratios (RR) at different time points during the dialysis sessions for Trp, IS, IAA and urea calculated from the dialysate values (*n* = 244). *** *p* < 0.001; ** *p* < 0.01; * *p* < 0.05 vs. previous timepoint value.

**Figure 4 ijms-21-01522-f004:**
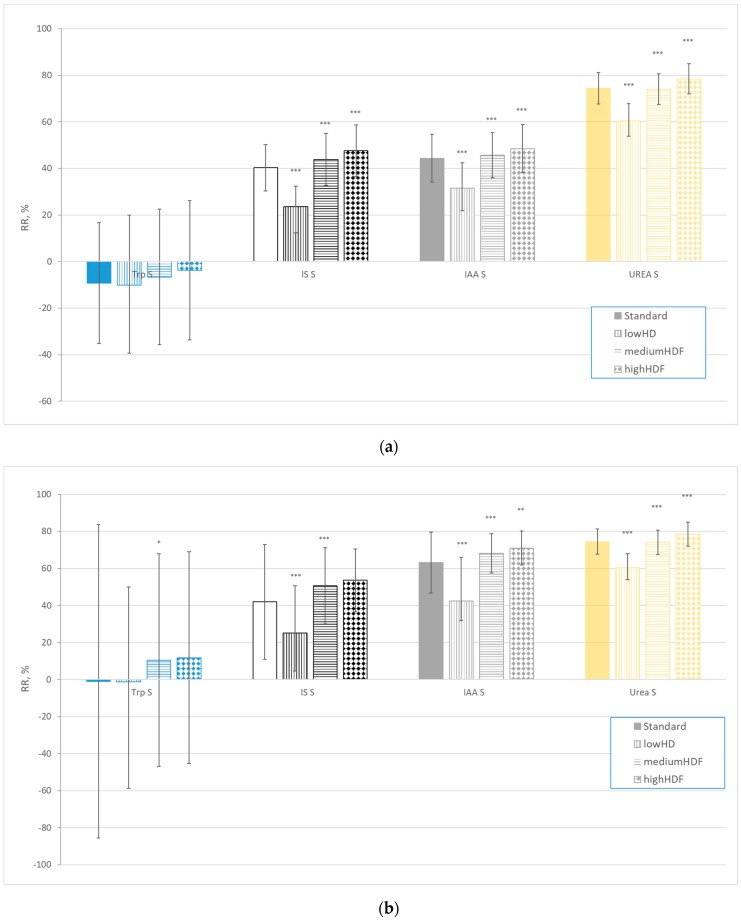
Reduction ratios (RR) and total removal of solute (TRS) calculated for urea, Trp, IS and IAA from the start and end (240 min) of the dialysis session for different dialysis modalities and settings: (**a**) RRs assessed from total serum concentrations (*n* = 77); (**b**) RRs assessed from free serum concentrations (*n* = 77); (**c**) RRs assessed from spent dialysate concentrations (*n* = 53); and (**d**) TRS of urea (in mmol), Trp, IS and IAA (in µmol) (*n* = 74). *** *p* < 0.001; ** *p* < 0.01; * *p* < 0.05 vs. previous modality value.

**Figure 5 ijms-21-01522-f005:**
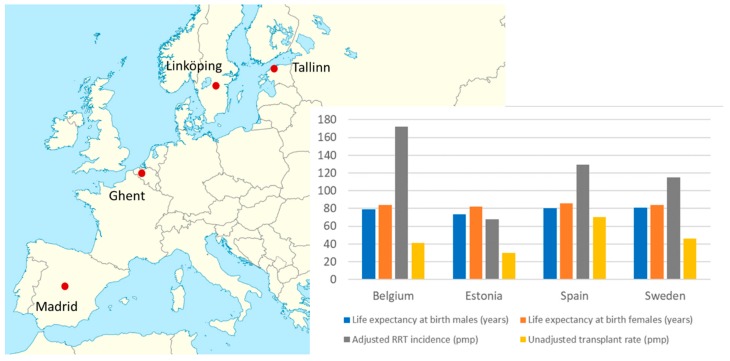
Location of centres and country life expectancy (GBD 2017 study) [57], renal replacement therapy (RRT) incidence and transplant rate (data from ERA-EDTA Registry, Belgium data correspond to Dutch-speaking Belgium) [58].

**Figure 6 ijms-21-01522-f006:**
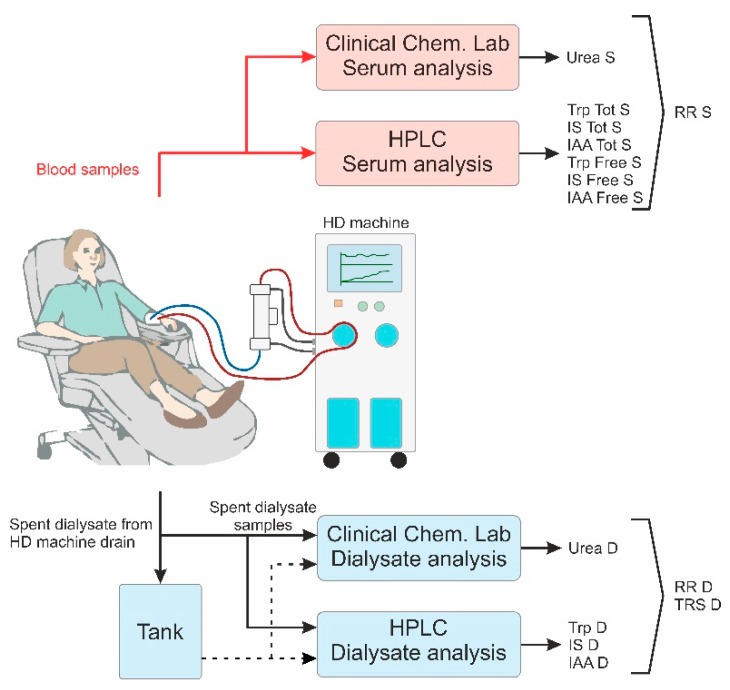
The schematic clinical set-up, sample collection, and analysis during the clinical studies.

**Table 1 ijms-21-01522-t001:** Mean pre-dialysis serum solute concentrations and standard deviations of patients by centre (in µmol/L, except for urea in mmol/L) ^1^.

	Solutes	All (*n* = 78)	Centre 1 (*n* = 22)	Centre 2 (*n* = 21)	Centre 3 (*n* = 15)	Centre 4 (*n* = 20)
**Total**	**Trp**	29.3 ± 8.0	29.00 ± 6.28	32.6 ± 9.7 **	27.7 ± 8.8 °°	27.3 ± 6.0 °°°
	**IS**	107.6 ± 51.2	94.8 ± 47.6	121.4 ± 57.0 **	98.5 ± 55.5 °	113.6 ± 40.0 **
	**IAA**	11.8 ± 9.2	13.5 ± 13.8	11.4 ± 5.9	11.3 ± 7.3	10.9 ± 6.7
**Free**	**Trp**	6.41 ± 2.07	6.22 ± 2.02	6.30 ± 1.63	5.54 ± 1.57 */°°	7.39 ± 2.50 **/°°/^§^
	**IS**	14.7 ± 8.7	11.3 ± 5.8	15.0 ± 9.1	12.7 ± 7.8	19.4 ± 9.4 ***/°°/^§^
	**IAA**	2.91 ± 2.00	3.08 ± 2.55	2.72 ± 1.68	2.61 ± 1.30	3.18 ± 2.06
	**Urea**	19.1 ± 5.9	21.2 ± 7.3	16.8 ± 4.1 ***	18.7 ± 6.5 */°	19.6 ± 4.4 °°°

^1^ The statistical differences are marked as *** *p* < 0.001; ** *p* < 0.01; * *p* < 0.05 vs. Centre 1; °°° *p* < 0.001; °° *p* < 0.01; ° *p* < 0.05 vs. Centre 2; ^§^
*p* < 0.001 vs. Centre 3.

**Table 2 ijms-21-01522-t002:** Clinical data of the CKD patients. Numerical values are given as number of patients in parentheses or as mean ± SD.

Patients	All (*n* = 78)	Centre 1 (*n* = 22)	Centre 2 (*n* = 21)	Centre 3 (*n* = 15)	Centre 4 (*n* = 20)
Diagnosis ^1^	ADPKD (8); Diabetes (13); GN (16); Hypertension (12); Other (10); Renal; carcinoma (4); TIN (8); Unknown (7)	Diabetes (4); Hypertension (8); GN (3); TIN (3); Other (2); Renal carcinoma (2)	ADPKD (4); Diabetes (3); GN (4); Hypertension (2); Other (1); Renal carcinoma (2); TIN (1); Unknown (4)	ADPKD (2); Diabetes (2); GN (5); Hypertension (1); Other (4); TIN (1)	ADPKD (2); Diabetes (4); GN (4); Hypertension (1); Other (3); TIN (3); Unknown (3)
Age (years)	63 ± 16	55 ± 17	71 ± 11	59 ± 15	68 ± 14
Gender	M (63), F (15)	M (16), F (6)	M (16), F (5)	M (13), F (2)	M (15), F(5)
Race, Caucasian (%)	94	100	90	93	90
BMI, kg/m^2^	26.5 ± 5.5	26.8 ± 5.8	26.5 ± 3.7	26.2 ± 7.1	26.4 ± 5.4
BW, kg	78.3 ± 18.3	81.5 ± 21.3)	77.1 ± 12.6	81.4 ± 19.7	73.8 ± 17.5
Ultrafiltration volume, mL	2250 ± 1049	2565 ± 1190	1746 ± 1031	2208 ± 807	2468 ± 833
Residual diuresis, n (%)	30 (38%)	8 (35%)	14 (67%)	8 (53%)	0
Urinary volume, mL	342 ± 607	227 ± 397	700 ± 754	457 ± 713	0
Serum total protein, g/L	65.1 ± 5.8	62.8 ± 5.5	68.5 ± 4.6	65.2 ± 5.5	63.8 ± 5.9
spKt/V_urea_	1.64 ± 0.34	1.48 ± 0.30	1.70 ± 0.23	1.51 ± 0.29	1.85 ± 0.37
Dialysis access	native fistula (64); graft (11); catheter (3)	native fistula (15); graft (7)	native fistula (19); graft (2)	native fistula (12); catheter (3)	native fistula (18); graft (2)
Dialysis vintage, months	55 ± 66	50 ± 50	86 ± 04	47 ± 31	35 ± 23

^1^ ADPKD: autosomal dominant polycystic kidney disease; GN: glomerulonephritis; TIN: Tubulointerstitial nephritis; unknown: chronic kidney disease of unknown aetiology; M: male; F: female.

**Table 3 ijms-21-01522-t003:** Dialysis treatment settings.

	Standard	LowHD	MediumHDF	HighHDF
Modality	HD/HDF	HD	HDF	HDF
Vs, L	23.0 ± 3.5	0	16.4 ± 3.0	24.6 ± 4.0
Time, min	240	240	240	240
Qb, mL/min	323 ± 40	200 ± 10	306 ± 62	378 ± 30
Qd, mL/min	458 ± 63	301 ± 11	793 ± 57	793 ± 47
Filter area ^1^, m^2^	1.90 ± 0.20	1.62 ± 0.19	2.13 ± 0.13	2.13 ± 0.14
Number of dialyses, n	78	78	78	78

^1^ Dialysis 1: FX60 (*n* = 6), FX80 (*n* = 6), FX800 (*n* = 30), FX1000 (*n* = 12), Polyflux210H (*n* = 12), Solacea 19H (*n* = 1), Elisio-19H (*n* = 9). Dialysis 2: Lo15 (*n* = 20), FX60 (*n* = 37), FX1000 (*n* = 1), Revaclear300 (*n* = 19), Solacea 15H (*n* = 1). Dialysis 3: FX800 (*n* =9), FX1000 (*n* =56), Polyflux210H (*n* =12), Solacea 21H (*n* =1). Dialysis 4: FX800 (*n* =11), FX1000 (*n* =56), Solacea 21H (*n* =1), Polyflux210H (*n* =10). The effective membrane area of dialyzers were the following: FX8 1.4 m^2^, FX60 1.4 m^2^, FX80 1.8 m^2^, FX800 1.8 m^2^, FX1000 2.2 m^2^, Solacea 15H 1.5 m^2^, Solacea 19H 1.9 m^2^, Solacea 21H 1.9 m^2^, Elisio 19H 1.9 m^2^, Polyflux 210H 2.1 m^2^, Revaclear 300 1.4 m^2^, Xevonta LO 15 1.5 m^2^.

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
