# Peer review of "Serum Levels and Removal by Haemodialysis and Haemodiafiltration of Tryptophan-Derived Uremic Toxins in ESKD Patients"

_ijms, 2020, doi:10.3390/ijms21041522_

Round 1
Reviewer 1 Report
This prospective multicentre study in four European dialysis centres ijms-725919 “Serum Levels and Removal by Haemodialysis and Haemodiafiltration of Tryptophan-Derived Uremic 3 Toxins in ESKD Patients” has relevance to the audience of this journal.
The aim of the present study was to evaluate serum levels and removal during haemodialysis and haemodiafiltration of tryptophan and tryptophan-derived uremic toxins, indoxyl sulfate (IS) and indole acetic acid (IAA), in ESKD patients in different dialysis treatment settings.
The authors report that Conventional low-flux dialysis may not adequately clear tryptophan-related uremic toxins while high efficiency haemodiafiltration increased tryptophan losses.
Comments
The present study investigates a hot and interesting issue: serum levels and removal during haemodialysis and haemodiafiltration of tryptophan and tryptophan-derived uremic toxins. It is well written; the text, the tables, and the figures are suitable and informative. References are up to date. There are practical implications of the results of this study in the effort to improve removal during haemodialysis or haemodiafiltration in patients with ESRD.Author Response
The following changes have been made to the revised version of manuscript.
All the changes are trackable except for new added citations for the disccusions section.
Numerical values have been added to the abstract for concentration change and RR.
Tryptophan has been abbreviated to Trp throughout article.
Figure 1 a has been modified: secondary axis has been added and values of IAA and urea have been scaled with the secondary axis to differentiate concentration changes more clearly.
The statement in the text about serum free IAA concentration rise during the first hour of the dialysis sessions compared to the initial pre-dialysis levels has been corrected throughout the article. Correct statement is that serum free IAA concentration decrease 0.8% in comparison to the initial pre-dialysis level.
Table 3. Residual diuresis values for 2 patients in centre 3 have been corrected.
Discussion has been supplemented with the text about free fatty acid displacer effect and the overall amino acid behaviour of other amino acids during haemodialysis and haemodiafiltration. Some new citations have been added regarding this. Typo in citation 13 has been corrected.
Some other minor spelling mistakes have been corrected.
Some details have been changed in authors affiliation section to the following:
line 13: 2 Centre of Nephrology, North Estonia Medical Centre, Tallinn, Estonia;
line 16: 3 Nephrology Division, Ghent University Hospital, Ghent, Belgium;
line 18: 4 Department of Nephrology and Department of Medicine and Health Science, Linköping University, Linköping, Sweden;
line 22: first name of Didider Sanchez-Ospina has been abbreviated to D.
Reviewer 2 Report
The authors reported serum levels and removal by dialysis treatment of tryptophan, indoxyl sulfate, and indole acetic acid. This is the first report to understand the change of tryptophan during dialysis treatment.
- The authors should report actual data of serum level or reduction rate of uremic toxins in the Abstract.
- Figure 1 is difficult for readers to understand the removal of IAA and urea, and the authors should show the date of each uremic toxin, separately with each scale of concentration.
- The authors focused the study about tryptophan-derived uremic toxins, however, they should discuss other amino acids in the Discussion.
Author Response
All of the changes are traceable except for new added citations for the discussion section.
Point 1: The authors should report actual data of serum level or reduction rate of uremic toxins in the Abstract.
Response 1: Numerical values have been added to the abstract for concentration change and RR.
Point 2: Figure 1 is difficult for readers to understand the removal of IAA and urea, and the authors should show the date of each uremic toxin, separately with each scale of concentration.
Response 2: Figure 1 a has been modified: secondary axis has been added and values of IAA and urea have been scaled with the secondary axis to differentiate concentration changes more clearly.
Point 3: The authors focused the study about tryptophan-derived uremic toxins, however, they should discuss other amino acids in the Discussion.
Response 3: Discussion has been supplemented with the text about free fatty acid displacer effect and the overall amino acid behaviour of other amino acids during haemodialysis and haemodiafiltration. Some new citations have been added regarding this. Typo in citation 13 has been corrected.
Other changes that were made are following.
Tryptophan has been abbreviated to Trp throughout article.
The statement in the text about serum free IAA concentration rise during the first hour of the dialysis sessions compared to the initial pre-dialysis levels has been corrected throughout the article. Correct statement is that serum free IAA concentration decrease 0.8% in comparison to the initial pre-dialysis level.
Table 3. Residual diuresis values for 2 patients in centre 3 have been corrected.
Some other minor spelling mistakes have been corrected.
Some details have been changed in authors affiliation section to the following:
line 13: 2 Centre of Nephrology, North Estonia Medical Centre, Tallinn, Estonia;
line 16: 3 Nephrology Division, Ghent University Hospital, Ghent, Belgium;
line 18: 4 Department of Nephrology and Department of Medicine and Health Science, Linköping University, Linköping, Sweden;
line 22: first name of Didider Sanchez-Ospina has been abbreviated to D.